# Neighborhood Socioeconomic Resources and Crime-Related Psychosocial Hazards, Stroke Risk, and Cognition in Older Adults

**DOI:** 10.3390/ijerph18105122

**Published:** 2021-05-12

**Authors:** Linda D. Ruiz, Molly Brown, Yan Li, Elizabeth A. Boots, Lisa L. Barnes, Leonard Jason, Shannon Zenk, Philippa Clarke, Melissa Lamar

**Affiliations:** 1College of Science and Health, DePaul University, Chicago, IL 60604, USA; lindadruiz@gmail.com (L.D.R.); molly.brown@depaul.edu (M.B.); YLI34@depaul.edu (Y.L.); ljason@depaul.edu (L.J.); 2Department of Psychology, University of Illinois at Chicago, Chicago, IL 60607, USA; eboots2@uic.edu; 3Rush Alzheimer’s Disease Center, Rush University Medical Center, Chicago, IL 60612, USA; lisa_l_barnes@rush.edu; 4Department of Psychiatry and Behavioral Sciences, Rush University Medical Center, Chicago, IL 60612, USA; 5Department of Health Systems Sciences, University of Illinois at Chicago, Chicago, IL 60612, USA; shannonzenk@me.com; 6Institute for Social Research, University of Michigan, Ann Arbor, MI 48104, USA; pjclarke@umich.edu

**Keywords:** crime, cardiovascular risk, attention, information processing, aging

## Abstract

Living in neighborhoods with lower incomes, lower education/occupational levels, and/or higher crime increases one’s risk of developing chronic health problems including cardiovascular disease risk factors and stroke. These cardiovascular health problems are known to contribute to cognitive decline and dementia. The purpose of this study was to determine the association of neighborhood socioeconomic resources and crime-related psychosocial hazards on stroke risk and cognition, hypothesizing that cardiovascular health would mediate any relationship between the neighborhood-level environment and cognition. The study evaluated 121 non-demented Chicago-area adults (~67 years; 40% non-Latino White) for cardiovascular health problems using the Framingham Stroke Risk Profile 10-year risk of stroke (FSRP-10). The cognitive domains that were tested included memory, executive functioning, and attention/information processing. Neighborhood socioeconomic resources were quantified at the census tract level (income, education, and occupation); crime-related psychosocial hazards were quantified at the point level. Structural equation modeling (SEM) did not show that the FSRP-10 mediated the relationship between neighborhood characteristics and domain-specific cognition. The SEM results did suggest that higher crime rates were associated with a higher FSRP-10 (*β*(105) = 2.38, *p* = 0.03) and that higher FSRP-10 is associated with reduced attention/information processing performance (*β*(105) = −0.04, *p* = 0.02) after accounting for neighborhood socioeconomic resources. Clinicians may wish to query not only individual but also neighborhood-level health when considering cognition.

## 1. Introduction

Increasingly, studies show that neighborhood-level factors of the social and built environment are associated with a broad range of individual-level outcomes among older adults [1]. For example, there is strong evidence of geospatial patterns in physical (e.g., cardiovascular disease risk factors) and behavioral (e.g., cognition) health based on the physical, social, and socioeconomic characteristics of a neighborhood [2,3]. While most research has focused on the physical characteristics of the neighborhood environment (i.e., walkability, transportation, and aesthetics) [1], there is evidence to suggest that the state of the socioeconomic neighborhood environment also has a significant impact on health outcomes [4,5]. Specifically, individuals living in the most disadvantaged neighborhoods based on lower income, education, and/or occupation levels are at higher risk for developing coronary heart disease [6] and other adverse cardiovascular outcomes [7], associations that persist even after adjusting for individual-level socioeconomic status and race/ethnicity [6,8,9].

Neighborhood-level socioeconomic disadvantage based not only on lower income, education, and/or occupation levels but also on psychosocial hazards including social disorganization and public safety concerns are also associated with increased rates of cognitive decline and lower cognitive performance more generally, even after controlling for individual-level socioeconomic status (SES) [2]. Furthermore, high rates of neighborhood-level violent crime is associated with worse performance on a cognitive screening method commonly used in older adults [8]. One potential mechanism to explain the impact of neighborhood-level disadvantage on individual-level cognition is through health, e.g., through increased cardiovascular disease and associated risk factors [9,10]. There is little research, however, that considers all of these factors simultaneously.

The aim of the present cross-sectional study was to determine the relationships between neighborhood-level socioeconomic disadvantages and individual-level health and cognition in non-demented community-dwelling older adults. Based on the literature linking neighborhood socioeconomic disadvantage [4,5,11,12] including safety [13] to cardiovascular health, and cardiovascular health to cognition [9,10], we hypothesized that individual-level cardiovascular health (i.e., 10-year risk of stroke) would mediate associations between neighborhood-level socioeconomic (i.e., lower income, education, and occupation levels) and psychosocial (i.e., violent crime rates) hazards and individual-level cognition. While crime statistics have not traditionally been included as indices of neighborhood-level disadvantage, their use is increasing [8] due, in part, to a recent systematic review citing crime as a key consideration when investigating the impact of neighborhood environments on cognition in older adults [2]. When taken together, our study demonstrates that neighborhood-level crime is significantly associated with individual-level stroke risk, which, in turn, impacts individual-level cognition independent of other neighborhood-level socioeconomic resources.

## 2. Materials and Methods

This study was funded by the National Institute on Aging to investigate cardiovascular disease risk factors, neighborhood ‘health’ factors, cognition, and brain aging. The study was approved by the University of Illinois at Chicago Institutional Review Board (UIC IRB #2012-0142) as well as the Rush University Medical Center IRB (#16102101). It was conducted in accordance with the Declaration of Helsinki with written informed consent obtained from all participants.

### 2.1. Participants

Described in detail elsewhere [14,15,16], individuals aged 60 or older from one of three self-identified ethnic/racial categories (i.e., non-Latino White or Black, and Latino) were recruited via community outreach and word of mouth for a study of healthy brain aging conducted between 2012 and 2016. An initial telephone screening conducted in participants’ language of choice (English or Spanish) determined study eligibility. At this screening, exclusion criteria consisted of a positive self-report of any of the following: a current or past history of neurological conditions including Alzheimer’s disease and related dementias, Parkinson’s disease or any other movement disorder; prevalent stroke; a current or past history of psychiatric disorders (e.g., depression or bipolar disorder); a history of head injury or loss of consciousness; a present or past history of substance abuse or dependence; psychotropic medication use or contraindications for magnetic resonance imaging (MRI) including metallic implants, cardiac pacemaker/defibrillator, and claustrophobia. A self-reported history of stable (e.g., diabetes) or remitted (e.g., cancer) medical illness was not an exclusionary factor. Individuals were not eligible if they had received cognitive testing within the past year or if they reported current involvement in a study with cognitive testing.

Following successful completion of the telephone screening, eligible individuals were scheduled for a more detailed in-person screening that included the Mini-Mental State Examination (MMSE) [17] and the Structured Clinical Interview for DSM-IV-TR (SCID) [18] for final inclusion and exclusion determination. These measures were administered by a trained research assistant (i.e., either a clinical psychology graduate student or Master’s level social worker) formally trained on SCID administration and fluent in either English or Spanish and were followed by an evaluation by a psychiatrist who completed the 17-item Hamilton Depression Rating Scale (HAM-D) [19]. All raters were blind to the telephone screening information. Final inclusion criteria consisted of an MMSE score ≥ 24, an absence of a psychiatric symptoms based on the SCID, a score ≤ 8 on HAM-D, and a lack of subjective memory complaints.

One-hundred and twenty-one participants met all inclusion and exclusion criteria and were enrolled in the study. We excluded 10 participants who were administered Spanish versions of the cognitive measures given concerns about comparability of select test measures used in the current analyses, and 6 individuals who either lacked information on key variables in our analyses or evidenced incidental findings on MRI. Thus, 106 participants contributed to the current analyses.

### 2.2. Neighborhood-Level Socioeconomic Environment

Participants provided their current address and duration of residence at their current address. If the stated duration of residence was less than 5 years (*n* = 21), participants were asked to provide their immediately prior address and duration of residence at that location for geocoding.

Geographic information systems (GIS) coding was used to analyze participants’ addresses. Data on neighborhood-level income, education, and employment were collected at the census tract level from the American Community Survey (2010 to 2014). Neighborhood-level crime data were collected at the point level (using a 1-mile buffer) from the Chicago Police Department’s Citizen Law Enforcement Analysis and Reporting (CLEAR) database accessed in 2015. Given that available GIS data were not in perfect one-to-one correspondence with all study visit years, we time lagged GIS data to 2 years prior to a participant’s year of study visit to maximize our sample. Thus, the resulting GIS data represented the socioeconomic environment captured 2 years prior to a participant’s study visit.

In accordance with methods previously outlined [20], a standardized index of neighborhood-level socioeconomic resources was constructed based on variables representing the following domains: income (variables included percent of the population with income below poverty level and median household income), occupation (variable included percent of the eligible civilian workforce population classified as unemployed), and education (variables included percent of population with less than 12 years of education and percent of the population with more than 16 years of education). These five variables were subjected to a principal component analysis (PCA), and the first principal component, which accounts for the largest proportion of total variance in any unrotated PCA, was retained. As seen in Table 1, the resulting composite score represented the individual standardized weighted coefficients of all five variables. Relevant items were recoded to ensure that lower composite scores indicated lower socioeconomic resources.

Crime variables representing per capita rates of homicide, robbery, and assault were quantified and combined using a similar PCA procedure as that described above. This allowed for the construction of a composite score that accounted for individual standardized weighted coefficients of these 3 variables (Table 2). Higher values on this composite reflect higher levels of violent crime.

### 2.3. Cardiovascular Disease Risk

Participants received a medical history interview and physical examination conducted by trained staff and a registered nurse, respectively, from the UIC Clinical Research Center (CRC). This evaluation included two seated blood pressure measurements separated by 5 min, anthropometrics including height and weight, and a confirmed 12 h fasting blood draw for health-related variables such as glucose and hemoglobin A1c. An electrocardiogram and medication review were also performed. This evaluation allowed for an assessment of the revised Framingham Stroke Risk Profile score 10-year risk of stroke (FSRP-10) [21]. The FSRP-10 (a higher score indicates a higher risk) was derived from the following cardiovascular disease risk factors: systolic blood pressure, antihypertensive medication use, diabetes mellitus, diabetes medication, current cigarette smoking, cardiovascular disease, and atrial fibrillation.

### 2.4. Cognition

Participants underwent a comprehensive neuropsychological assessment conducted by trained research assistants. We statistically grouped select cognitive test measures shown to be particularly vulnerable to increased cardiovascular disease and stroke risk in older adults (e.g., [22]). As detailed elsewhere [14], PCA procedures including varimax rotation resulted in the following three domains: (a) verbal learning, memory, and recognition (LMR); (b) attention and information processing (AIP); and (c) executive functioning (EF).

The LMR domain was based on total recall across five consecutive learning trials, long delay free recall, and a recognition discriminability index from The California Verbal Learning Test-II (CVLT-II) [23]. The AIP domain consisted of time to completion for Trail Making Test (TMT) Part A and Motor Trails as well as the Digit Symbol Coding subtest score from the Wechsler Adult Intelligence Scale, Fourth Edition (WAIS-IV). Lastly, the EF domain included TMT Part B minus Part A; the Matrix Reasoning subtest score from the Wechsler Abbreviated Scale of Intelligence, Second Edition (WASI-II); the WAIS-IV Letter–Number Sequencing subtest score; and total output during letter fluency (F, A, and S). We created continuous composite measures of these three cognitive domains by averaging z-scores for test items comprising each domain. Relevant test scores were recoded to ensure that higher values equated to worse performance. Cronbach’s alpha confirmed the PCA results for each cognitive domain (LMR = 0.89, AIP = 0.64, and EF = 0.73).

### 2.5. Statistical Analyses

Descriptive characteristics of our sample and initial bivariate correlations of key variables of interest were conducted using SPSS Version 27. Structural equation modeling (SEM) was employed using Mplus (Version 8) to evaluate whether stroke risk mediates relationships between neighborhood socioeconomic characteristics and cognition. Specifically, we tested three models examining stroke risk as a mediator of any association between the neighborhood-level PCA scores of socioeconomics and crime with cognition, i.e., LMR, AIP, and EF composite scores (separately). In order to assess the extent to which the model fit the data, the chi-squared (*X*^2^) statistic and several practical fit indices were utilized to evaluate the model, including the root mean square error of approximation (RMSEA), the comparative fit index (CFI), and the Tucker–Lewis index (TLI). While the chi-squared test is sensitive to sample size bias, it is considered an adequate metric for samples between 75 and 200 with suggested cutoff values greater than *p* = 0.05 representing better fit [24]. RMSEA is less influenced by large sample sizes with suggested cutoff values of 0.01, 0.05, and 0.08 indicating excellent, good, and mediocre fit, respectively [25]. CFI values approaching 1 and TLI values over 0.90 are indicative of acceptable fit [26].

## 3. Results

### 3.1. Participants

Participants included in these analyses were on average 67 years of age, equally distributed between female/male sex, racially and ethnically diverse (only 42.9% non-Latino White), and obtained an average of 16 years of education. Additional participant characteristics are outlined in Table 3.

As may be seen from Figure 1, despite our low sample size, we nonetheless had a good range of the 77 community areas (i.e., neighborhoods) throughout Chicago and the larger Cook County area from which to evaluate the socioeconomic and psychosocial hazards.

### 3.2. Correlations of Key Variables of Interest

A bivariate Pearson correlation matrix was created using the following variables of interest: neighborhood-level socioeconomic- and crime-related composites, as well as individual-level stroke risk (FSRP-10) and cognition (LMR, AIP, and EF composite scores). Of the correlations between the two neighborhood-level composites and individual-level stroke and cognitive status variables, only the relationship between the socioeconomic and crime composite scores were significant, r(105) = −0.65, *p* < 0.001. Thus, lower neighborhood-level socioeconomic resources were associated with higher neighborhood-level crime. The entire correlation matrix is outlined in Table 4.

### 3.3. Structural Equation Modeling

Initial models testing whether stroke risk mediated the relation between predictors (neighborhood-level socioeconomic resources and crime) and outcome variables (LMR, AIP, and EF composite scores, separately) run with all paths freely estimated demonstrated that the direct effects between predictor and outcome variables were nonsignificant (data not shown). Thus, these direct effects were constrained to zero for final models outlined below.

#### 3.3.1. Learning, Memory, and Recognition (LMR)

While model fit was adequate for using LMR (*X*^2^ (12, *N* = 106) = 1.81, *p* = 0.41; *RMSEA* < 0.01; *CFI* = 1; *TLI* = 1.25), FSRP-10 was not significantly associated with the LMR composite score, *β*(106) = −0.004, *p* = 0.86.

#### 3.3.2. Attention/Information Processing (AIP)

All but one model fit statistic was adequate for the AIP model (*X*^2^ (12, *N* = 105) = 2.62, *p* = 0.27; *RMSEA* = 0.05; *CFI* = 0.91; *TLI* = 0.78); despite TLI results, FSRP-10 was significantly correlated with AIP (*β*(105) = 0.04, *p* = 0.03). Specifically, higher 10-year stroke risk was associated with higher values (equating to worse performance) on attention/information processing tasks. Thus, we felt that the overall model fit was acceptable. Additionally, neighborhood socioeconomic characteristics, in terms of socioeconomic resources (*β*(105) = 2.38, *p* = 0.04) as well as crime-related psychosocial hazards (*β*(105) = 1.26, *p* = 0.02), were significantly correlated with FSRP-10. The final model is described graphically in Figure 2.

#### 3.3.3. Executive Functioning (EF)

The EF model had poor overall fit (*X*^2^ (12, *N* = 102) = 3.53, *p* = 0.17; *RMSEA* = 0.09; *CFI* = 0.74; *TLI* = 0.36) and was, therefore, not considered further.

## 4. Discussion

In this cross-sectional study of just over 100 older adults, the results revealed that higher neighborhood-level crime rates were associated with higher stroke risk and that a higher stroke risk was associated with higher values (equating to worse performance) on attention/information processing tasks. Similar investigations of other cognitive domains including learning and memory as well as executive functioning did not suggest a similar results profile. Thus, while stroke risk may not formally mediate the relationship between neighborhood-level characteristics and domain-specific and individual-level cognitive skills, higher crime rates were associated with stroke risk and a higher stroke risk was associated with reduced cognitive performance after accounting for other neighborhood socioeconomic characteristics. Furthermore, our study suggests that where one lives is associated with how one lives both physically and, in turn, cognitively, particularly as it relates to attention and information processing.

The results of this study contribute to the literature in several ways. First, this study represents a growing body of literature that aims to draw connections between neighborhood-level factors and individual-level health outcomes [2,27], extending this work to include the fact that geographic location matters for stroke risk and that stroke risk matters for cognitive functioning. Second`, several recent reviews of the literature regarding the neighborhood environment and cognition in older adults advocated for more work studying mediators to elucidate the underlying mechanisms linking neighborhood-level factors and cognition [2,8]. The results of this empirically-based study also highlight the need to investigate the interplay between neighborhood-level crime and individual-level stroke risk as it may contribute to cognitive functioning in older community-dwelling adults.

While the underlying biological mediators of the associations revealed in this study cannot be gleaned directly from the work presented, suggestions for future investigations incorporating biological biomarkers may be found in the literature. For example, the literature suggests that chronic exposure to stressful environments has a negative impact on health, including an increased risk for developing cardiovascular disease risk factors such as hypertension, diabetes, and coronary heart disease, all of which are associated with increased risk for stroke and other adverse outcomes [28]. An underlying assumption of our work is that living in neighborhoods with higher crime is stressful. In fact, there is evidence that living in such environments is associated with greater cortisol dysregulation, which is a biomarker of increased stress [29]. While stress (or cortisol) is not directly measured in this study, our data support the assertion that stressful environments matter for individual-level health. In fact, their importance may extend beyond stroke risk such that dangerous environments may also be indirectly associated with cognitive health through stroke risk in community-dwelling older adults.

The assumption of chronic exposure to stressful environments may not, however, apply to all neighborhood-level characteristics of the environment in our structural equation models. Specifically, the significant association between our measure of socioeconomic resources and stroke risk was in opposition to our hypotheses as well as much of the literature. Thus, our association between lower socioeconomic resources and lower stroke risk contrasts previous work suggesting that living in more socioeconomically impoverished neighborhoods has an adverse effect on individual-level health [4,29]. Post hoc analyses, including simple correlations (see Appendix A), suggest that the socioeconomic resources driving our counterintuitive results were neighborhood-level educational attainment: i.e., individuals living in neighborhoods with high educational attainment had significantly higher stroke risk, and vice versa. A recent study examining the relationship between race, SES, and neuroimaging markers of structural brain integrity revealed that higher SES was associated with greater total brain, and gray and white matter volumes in non-Latino Whites but not in non-Latino Blacks [30]. These investigators hypothesized that differential exposure to contextual stressors including experiences of discrimination and/or altered cardiovascular responses to such stressors, particularly relevant for non-Latino Blacks with higher SES, may explain their results. Given our relatively limited sample size and a lack of data on select contextual factors, including experiences of discrimination, an exploration of racial differences in this study was not possible. Future research is needed at the intersection of race and place to determine whether disparities in contextual stressors unique to non-Latino Blacks and Latinos may have influenced the direction of the association between socioeconomic resources and stroke risk in our study.

The strengths of this paper include the fact that this is amongst the first study to examine the mediators between neighborhood-level socioeconomic environments including crime-related psychosocial hazards and cognition. Additional strengths include our analytic sample consisting of a diverse cohort of community-dwelling older adults without dementia and our focus on stroke risk, not prevalent or incident disease as related to neighborhood context and cognition. Additionally, we employed sophisticated statistical methodology and publicly available data on neighborhood-level health to understand the interplay between neighborhood- and individual-level health on cognition in older adults. Nonetheless, our work should be considered within the context of its limitations.

Noted limitations of our work include the cross-sectional nature of this study which does not allow for an understanding of causality. While our requirement that participants provide an address that denoted at least a five-year duration of exposure to their neighborhood environment does not counter our cross-sectional design, when combined with our findings that individuals are likely to live in socioeconomically similar regions throughout their life [31,32], the five-year requirement adds to our assumption that participants in this study had at least a reasonable duration of exposure to the neighborhood-level socioeconomic environments, including the crime-related psychosocial hazards investigated in the current research. Regardless of whether our findings represent an acute or more chronic effect of exposure, the results may nonetheless support the increasingly accepted idea that ecological risk factors have an impact on cardiovascular as well as cognitive health [33]. Additional study limitations include the relatively small sample size and the absence of sample size calculations, both of which may have contributed to a lack of power to detect effects of interest. Furthermore, participants evidenced a relatively low 10-year risk of stroke, suggesting this was a relatively healthy cohort. These study limitations may have negatively impacted our work, e.g., the relative health of the cohort may have introduced bias into the sample such that the average participant may or may not be representative of the population in their surrounding area. Nonetheless, the fact that we detected an effect suggests that future work in less healthy and/or larger populations may also reveal these associations.

## 5. Conclusions

This study demonstrated that higher amounts of neighborhood-level crime negatively associated with stroke risk and, in turn, this stroke risk associated with attention and information processing. While work is ongoing to clarify the role of socioeconomic resources, more specifically neighborhood-level educational attainment, on individual-level physical and cognitive health outcomes, our findings with neighborhood-level crime have clinical practice implications. Specifically, the neighborhood represents an important context that should be considered by clinicians as part of the diagnostic interview and case conceptualization process. While it may not be common for clinicians to specifically ask about the neighborhood in which their patients live, doing so may provide a wealth of information about daily, chronic stressors that have implications for symptom presentation and possibly even long-term cognition.

## Figures and Tables

**Figure 1 ijerph-18-05122-f001:**
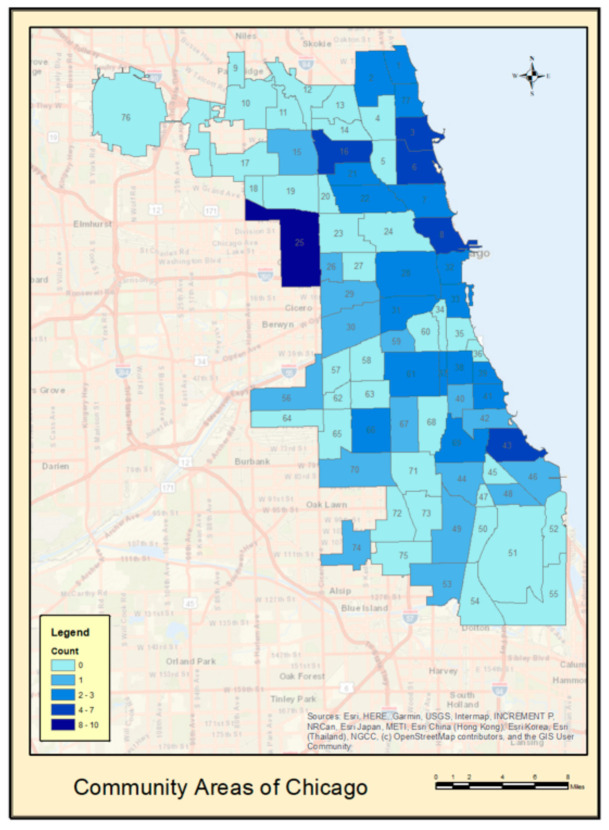
Neighborhood-level distribution of analytic sample.

**Figure 2 ijerph-18-05122-f002:**
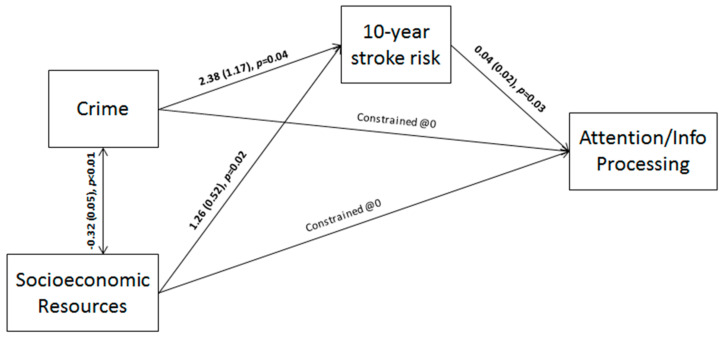
Final model outlining the relationships between the neighborhood socioeconomic environment, stroke risk, and attention/information processing.

**Table 1 ijerph-18-05122-t001:** Neighborhood-level socioeconomic-specific composite score.

	Unrotated Factor Loadings
% below poverty level	0.87
Median household income	−0.88
% with less than 12 years of education	0.79
% with 16+ years of education	−0.87
% unemployment	0.81
Total Variance Explained	71.83%

Abbreviations: *%* = percent.

**Table 2 ijerph-18-05122-t002:** Neighborhood-level crime-specific composite score.

	Unrotated Factor Loadings
Assault	0.99
Battery	0.95
Robbery	0.84
Sexual Assault	0.93
Homicide	0.90
Total Variance Explained	85.23%

Abbreviations: *%* = percent.

**Table 3 ijerph-18-05122-t003:** Participants characteristics (*N* = 106).

Age, M (SD)	67.91 (6.66)
Female, Sex *n* (%)	53 (50)
Race/Ethnicity, *n* (%)	
Non-Latino Black	50 (47.1)
Non-Latino White	48 (45.2)
Latino	8 (7.5)
Education, M (SD)	15.98 (2.86)
MMSE, M (SD)	28.66 (1.38)
FSRP-10, M (SD)	6.20 (4.86)

Abbreviations: M (SD) = mean (standard deviation); *n* (%) = number (percent); MMSE = Mini-Mental State Examination; FSRP-10 = Framingham Stroke Risk Profile score 10-year risk of stroke.

**Table 4 ijerph-18-05122-t004:** Correlation table between key variables of interest.

	1.	2.	3.	4.	5.
1. Socioeconomic composite	--				
2. Crime-related composite	**−0.65,**	--			
	***p* < 0.001**				
3. FSRP-10	0.08,	0.02,	--		
	*p* = 0.38	*p* = 0.82			
4. Cognition—LMR domain	0.09,	0.09,	−0.02,	--	
	*p* = 0.35	*p* = 0.34	*p* = 0.80		
5. Cognition—AIP domain	−0.05,	−0.13,	**0.22,**	**−0.33,**	--
	*p* = 0.58	*p* = 0.16	***p* = 0.02**	***p* < 0.001**	
6. Cognition—EF domain	0.09,	0.01,	−0.15,	**0.43,**	**−0.40,**
	*p* = 0.37	*p* = 0.88	*p* = 0.13	***p* < 0.001**	***p* < 0.001**

Abbreviations: FSRP-10 = FSRP-10 = Framingham Stroke Risk Profile score 10-year risk of stroke; LMR = verbal learning, memory, and recognition; AIP = attention and information processing; EF = executive functioning. Bolded entries signify significance at *p* < 0.05; shaded columns highlight correlations of interest.

## Data Availability

Data available on request due to restrictions both related to privacy and ethics.

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
