# Peer review of "Neighborhood Socioeconomic Resources and Crime-Related Psychosocial Hazards, Stroke Risk, and Cognition in Older Adults"

_ijerph, 2021, doi:10.3390/ijerph18105122_

Round 1

Reviewer 1 Report

In summary, in view of the gap in knowledge examining mediators of the relationships, the authors showed that higher neighbourhood-level crime was negatively associated with stroke risk, which in turn was associated with worse attention and information processing. The other proposed mediational pathways were not statistically fitting or did not reach statistical significance. 

Strengths: mediation analyses 

weaknesses: relatively small sample size for a SEM modelling and cross-sectional analyses impede causality inference

Introduction:

This section was well-described. 

Methods:

  1. the SCID was administered by a research assistant whereas the HAM-D was administered by a clinician. What was the rationale for this difference?
  2. No sample size calculation was presented. With just over 100 participants, the sample could well be underpowered, which could have rendered some of the mediations not fitting/significant.
  3. under "2.3 Cardiovascular disease risk", ...medical screen, history, and physical (examinations/ screenings)? there seem to be lacking a word there.

Results:

  1. Line 200- I would be more cautious to suggest that the sample represented the majority of the 50 wards as strongly it was suggested; one, due to the low sample size, and two, because of the low sample size, each ward was represented by very few participants. 
  2. it would be helpful to visualize the correlation matrix by presenting a table.
  3. Line 228: while the other fit statistics seemed acceptable, the TLI was lower than acceptable (<0.90), which was 0.78. Furthermore, this is the core mediation model for the paper, so attention would be placed on this model. 
  4. Description in text for the path from SES or crime to FSRP-10 was lacking.

Discussion:

  1. Lines 248-250: I am not sure what the message here is. This sentence needs modifications.
  2. Lines 278-281: the messages were there, but need re-writing for clarifications. One suggestion is to break the sentence into several sentences, which will help the readers to better grasp the messages.
  3. Lines 281-284: was this result presented in this paper in the result section? If not, to mention it here, it should be presented, at least  as a supplementary material 
  4. The sentences following line 284 suggest differential findings could be attributed to racial differences. However, this speculation did not seem to gel with what has been presented thus far. In the introduction, it would be useful to allude to this plausible difference. Furthermore, could interaction analyses be conducted to examine this postulated difference?
  5. Lines 295-300: While I understand that there are strengths to be highlighted, the ones mentioned were not really convincing strengths. Please highlight the fact that this is amongst the first study to examine the mediators between neighbourhood environment/ crime and cognition, which is the primary novelty of this paper. 
  6. For the limitation on cross-sectional design, the counter argument on the measures being chronic (5-year exposure) was not strong and did not make a strong case in counter-arguing this limitation. 
  7. Sample size limitation should be noted here as well, as indicated above.

Author Response

Pease see the attachment

Reviewer 2 Report

The MS presents a study that links distal socio-economic factors (resources, crime) with more proximal health and cognitive factors in a sample of older US persons. The study is operationally sophisticated and is clearly and succintly presented. The study, hence, has the capacity to inform this literature.

I only have two concerns/questions: Why in the main SEM model, the link between socioeconomic resources and the outcome (cognitive performance) is not measured but constrained. I would think the vibrant literature on socioeconomic inequality would speak to such an effect.

Were research assistants conducting the cognitive assessment cognizant of the research main aims? (see Rosenthal, 1994)

Rosenthal, R.(1994).Science and ethics in conducting analyzing and reporting psychologicalresearch. Psychol Sci, 5(3), 127‐134.

Reviewer 3 Report

1. Lines 69-70 seem more appropriate for discussion

2. Methods - describing as deprivation or disadvantage index makes directionality confusing- call it marker of socioeconomic resources

3. Figure 1 should not have dots representing those in study - risk of PII - highlight those wards that have participants so unknown how many people per ward

4. Why attention information processing and not other neurologic measures?

Round 2

Reviewer 1 Report

The authors have addressed all the concerns previously raised and I thus recommend this paper for publication.